# Research on Tuning Control Technology for Wireless Power Transfer Systems for Concrete Embedded Sensors

**Cancan Rong \*, Zhousen Wu, Lihui Yan, Mengmeng Chen, Jiaan Yan, Gang Ren and Chenyang Xia**

Jiangsu Province Laboratory of Mining Electric and Automation, China University of Mining and Technology, Xuzhou 221008, China; ts21130208p31@cumt.edu.cn (Z.W.); ts22230058a31@cumt.edu.cn (L.Y.); 17184992@cumt.edu.cn (M.C.); ts21130084a31ld@cumt.edu.cn (J.Y.); rengang@cumt.edu.cn (G.R.); chyxia@cumt.edu.cn (C.X.)
\* Correspondence: ccrong@cumt.edu.cn

**Abstract:** Concrete embedded sensors play a very important role in structural health monitoring. However, the time of endurance of sensors remains a performance bottleneck and sensors need to be charged without damaging the structure as well. Wireless power transfer (WPT) technology is a promising approach to solving this problem. However, the electromagnetic characteristics of concrete medium can cause WPT systems to be untuned and can reduce the energy transmission efficiency of the system. In this paper, the induced medium loss and eddy current loss of a WPT system in concrete are calculated using analytical equations and finite element analysis method. The equivalent circuit model of a concrete–air transmedia WPT system is established according to the calculated losses and a composite tuning control technology is proposed based on the above analysis. In addition, the composite tuning control technology combines the advantages of frequency-modulation tuning and dynamic compensation tuning to ensure the overall resonance of the WPT system. The tuning control technology can ensure the whole resonance of the WPT system and make the natural resonant frequencies of primary and secondary sides consistent. The experimental results show that compared with the untuned control technology, the output power and efficiency of the tuned control system increased by 73% and 11.05%, respectively. The proposed tuning control technology provides direction for future charging of concrete-embedded sensors.

**Keywords:** wireless power transfer; dielectric loss; eddy current loss; frequency-modulation tuning; dynamic compensation





## 1. Introduction

Concrete structures such as bridges, tunnels and buildings are slowly corroded and damaged due to the influence of environmental factors (wind, temperature, rain, earthquake, etc.), which seriously shortens the service life of buildings. In order to solve the potential safety hazards of concrete buildings, various sensors are used to obtain data about the interior of the building, such as water content, corrosion degree and so on. After analyzing the data, targeted maintenance of buildings with structural defects can be carried out to ensure their safety and reliability. However, the traditional wired sensor lines are complex and long-term exposure to harsh external environments will cause cable damage that affects the reliability of the sensors. More critically, cables embedded in a bridge will accelerate the internal seepage of the bridge and aggravate the internal corrosion of the bridge. Therefore, wireless sensors are preferred and are widely deployed in concrete structures to monitor the condition of structural health. However, traditional batteries are not a good choice for long-term deployments of concrete-embedded sensors. Obviously, wireless sensors can only stay operational for a very limited amount of time. In recent years, wireless power transfer (WPT) technology has become the focus of global research. Compared with wired charging, WPT systems have the advantages of free positioning, penetrating non-magnetic materials, safety, reliable power supply, etc.; they are widely

used in industry [1], transportation [2–6], drones [7], medicine [8–10] and power supply for underwater devices [11–15]. The application of WPT technology to the charging of embedded sensors in concrete buildings has great advantages. A structure diagram is shown in Figure 1.

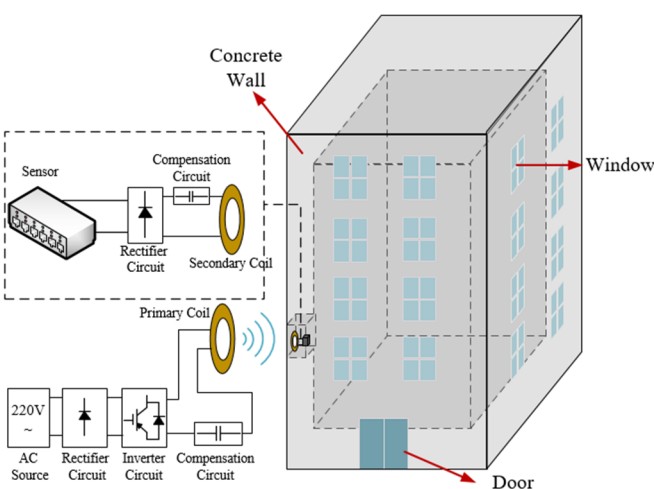

**Figure 1.** Air–concrete transmedia WPT system.

Embedded wireless sensors in concrete have been extensively studied by scholars around the world. The effects of three different media (air, unreinforced concrete and reinforced concrete), on the power transfer efficiency (PTE) of WPT systems were investigated in [16]. The results showed that the power transfer efficiency in concrete is 3% higher than in air, due to the increase in the magnetic induction strength by the paramagnetic material in the concrete. The influence of humidity of concrete materials on the WPT system in concrete media was studied in [17]. Humidity affects the conductivity and relative permittivity of concrete materials, thus causing additional loss to the WPT system. The experimental results show that radio energy transmission efficiency reaches 59% with a concrete humidity of 0.2% and 47% with a concrete humidity of 27%. A structural health monitoring system based on wireless power and data transfer (WPDT) method was proposed in [18]. The system can not only power the wireless sensor network, but also realize the bidirectional data transmission. The experimental results indicate that there is no distortion of data in either direction. Furthermore, this study demonstrates that bidirectional data transmission can be realized without additional circuits while performing wireless power transmission. In [19], a study on a high-power WPT system applied to rail transit was performed using analytical equations and finite element analysis (FEA) to analyze the various losses of concrete materials. The analytical and experimental results show that the eddy current loss of the steel reinforcement in the concrete is much larger than other losses when the system operating frequency is above 60 kHz. However, the influence of various losses on the circuit model parameters of the WPT system has not been analyzed in depth in this paper.

Compared with a WPT system placed in air, the air–concrete cross-medium WPT system will lead to changes in the self inductance and mutual inductance of the transmitting and receiving coils due to the influence of concrete; this can easily take the system out of resonance and can seriously reduce the transmission efficiency of the WPT system. The current research for WPT systems mainly focuses on optimizing transmission stability, optimizing transmission efficiency and improving transmission efficiency [20–23]. The two main research directions are parameter optimization design and tuning control of WPT system.

The tuning control of WPT systems has been studied extensively worldwide, and the main research directions are with the aim of adjusting the capacitance of resonant topology networks, adjusting the inductance of resonant topology networks and adjusting the working frequency of WPT systems. In [24], a circuit topology of three inductors in

series with adjustable capacitors and diodes in L-type series was proposed, which greatly improved the overall performance of the WPT system. In [25], a design for a tunable matching network based on dynamic load modulation of a tunable capacitor was presented. In [26], a neural network algorithm was combined with a capacitance matrix to identify the changes in the resonance parameters of a WPT system. However, the capacitance matrix cannot continuously adjust the equivalent capacitance value. This method is not suitable for systems with changing parameters such as inductance and mutual inductance of the coupling mechanism.

In [27], the resonant inductor was connected in series with the switch tube and the current in the inductor was adjusted by controlling the switch tube to change the equivalent inductance value. A magnetic amplifier was used instead of adjustable inductance in [28]. A frequency-modulation tuning strategy was proposed in [29] to improve the efficiency of a WPT system, aiming at the problem of resonance-network detuning of the system caused by complex underwater environment changes. Nevertheless, this method can only ensure that the WPT system maintains the resonance state as a whole and fails to guarantee simultaneous resonance in both the primary and secondary sides; the transmission performance of the system does not reach the optimal state. The comparison between this paper and previous research directions of scholars is shown in Table 1.

**Table 1.** Research direction.

| Research | Dielectric Loss | Eddy Current Loss | Magnetic Permeability | Efficiency Optimization |
|---|---|---|---|---|
| Reference [16] | × | × | √ | × |
| Reference [17] | × | √ | √ | × |
| Reference [19] | √ | √ | √ | × |
| Reference [30] | × | √ | × | √ |
| Reference [31] | × | √ | × | √ |
| This paper | √ | √ | √ | √ |

where "×" means that no relevant studies have been conducted and "√" means that that relevant research has been performed.

In this paper, the effects of the relative permittivity and conductivity of concrete on WPT systems are studied, and the equivalent circuit of an air–concrete transmedia WPT system is proposed. According to the detuning of WPT system caused by concrete, a composite tuning control strategy combining variable-capacitance dynamic compensation tuning and frequency-modulation tuning is adopted and experimentally verified. The experimental results show that the output power of the system is increased by 73% and the efficiency is improved by 11.05% with the tuning control.

## 2. Equivalent Circuit Model

### 2.1. Dielectric Loss Equivalent Circuit

There is often a large parasitic capacitance in the planar spiral coil. The influence of the parasitic capacitance can be neglected when the operating frequency of the system is very low. In the case of high frequency, the parasitic capacitance will have a great influence on the WPT system and can even move the WPT system out of the resonant state, reducing its power and efficiency. At the same time, the general load end must maintain a constant voltage output, which means that the inverter output end must provide a higher voltage, which poses a greater challenge to the design of the voltage stress of the power switch device.

A cross section of the planar spiral coil is shown in Figure 2. There is an insulating layer between the copper cores of the coil and the external medium of the coil. The external medium of the coil placed in the air is air, and the external medium of the planar spiral coil embedded in the concrete is concrete. The relative permittivity of the planar spiral coils placed in air and those embedded in concrete are different due to differences in the relative permittivity of air and concrete.

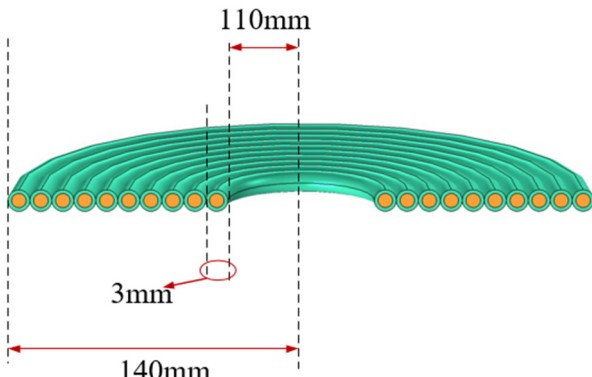

**Figure 2.** Cross-section diagram of a planar spiral coil.

In order to facilitate the calculation, the cross section of the planar spiral coil is simplified. The simplified cross section of the planar spiral coil is shown in Figure 3.

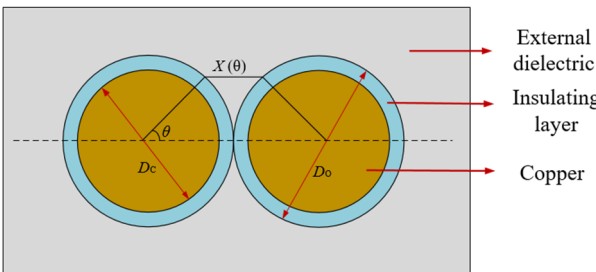

**Figure 3.** Simplified diagram of the cross section of a planar spiral coil.

In Figure 3, $Dc$ is the diameter of the copper-coil core (without considering the coil insulation), $Do$ is the diameter of the coil wire and $X(\theta)$ is the length of the electric field line. The differential expression of the parasitic capacitance between two adjacent wires can be obtained as [32]:

$$dC = \varepsilon \frac{dS}{x} \tag{1}$$

where $dS$ is the differential of the surface area of two adjacent wires, $\varepsilon$ is the relative permittivity of the medium and $x$ is the length of the electric field line on the surface of the wire. The interturn capacitance of the planar spiral coil is composed of two parts, one is the parasitic capacitance in the insulating layer and the other is the parasitic capacitance in the external medium of the insulating layer. The differential expression of the parasitic capacitance in the insulating layer is written as:

$$dC_t = \frac{\varepsilon_r \varepsilon_o}{dr} r d\theta dl \tag{2}$$

$$dC_t = \varepsilon_r \varepsilon_o d\theta \int_0^{l_t} dl \int_{r_c}^{r_o} \frac{r}{dr} = \frac{\varepsilon_r \varepsilon_o l_t}{\ln \frac{r_o}{r_c}} d\theta \tag{3}$$

where $r_c$ is the inner radius of the coil (excluding the insulating layer), $r_o$ is the outer radius of the coil (including the insulating layer) and $l_t$ is the length of the single-turn coil.

According to the geometric structure in Figure 3, the length of the electric field line can be obtained as follows:

$$x(\theta) = D_o(1 - cos\theta) \tag{4}$$

The surface area of the conductor per unit angle is expressed by the following equation:

$$dS = \frac{1}{2} D_o l_t d\theta \tag{5}$$

The differential expression of the parasitic capacitance in the external medium can be expressed as follows:

$$dC_g = \varepsilon_r' \varepsilon_o \frac{dS}{x(\theta)} = \varepsilon_r' \varepsilon_o \frac{l_t}{2(1 - \cos\theta)} d\theta \tag{6}$$

The total parasitic capacitance is obtained by connecting the parasitic capacitance of the insulating layer and the parasitic capacitance of the concrete medium in series:

$$dC_{eq} = \frac{\frac{1}{2}dC_t \cdot dC_g}{\frac{1}{2}dC_t + dC_g} = \frac{\varepsilon_r' \varepsilon_o l_t}{2} \frac{1}{1 - \cos\theta + \frac{\varepsilon_r'}{\varepsilon_r} \ln \frac{r_o}{r_c}} d\theta \tag{7}$$

$$C_{eq} = 2\varepsilon_r' \varepsilon_r \varepsilon_o l_t \frac{\arctan\sqrt{\frac{\varepsilon_r' \ln \frac{r_o}{r_c} + 2\varepsilon_r}{\varepsilon_r' \ln \frac{r_o}{r_c}}}}{\sqrt{(\varepsilon_r' \ln \frac{r_o}{r_c})^2 + 2\varepsilon_r' \varepsilon_r \ln \frac{r_o}{r_c}}} \tag{8}$$

where $\varepsilon_r$ is the relative permittivity of the coil insulation layer and $\varepsilon_r'$ is the relative permittivity of the external medium.

Also, the number of turns of the planar spiral coil affects the magnitude of the parasitic capacitance. The total parasitic capacitance of an N-turn helical coil can be equated to N-1 parasitic capacitors in series. The total parasitic capacitance is as follows:

$$C_S = \frac{C_{eq}}{n - 1} \tag{9}$$

The equivalent circuit of the single-turn coil considering the parasitic capacitance is shown in Figure 4. The impedance magnitude can be obtained as follows:

$$Z = (j\omega L + R_{ac}) \parallel \frac{1}{j\omega C_S} \tag{10}$$

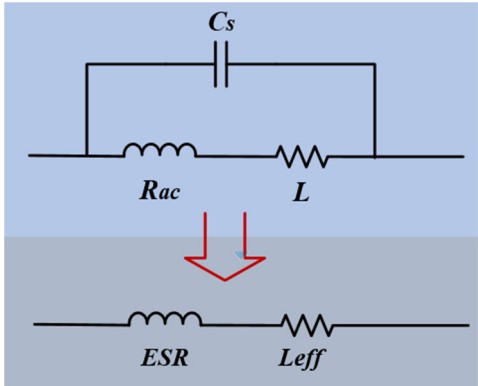

**Figure 4.** Single-turn coil equivalent circuit model.

A simplification can be obtained as follows:

$$Z \approx \frac{R_{ac}}{(1 - \omega^2 L C_S)^2} + j\omega \frac{L}{1 - \omega^2 L C_S} \tag{11}$$

Therefore, the planar spiral coil considering parasitic capacitance can be equivalent to a series circuit of inductance and resistance. The equivalent series impedance and equivalent inductance can be expressed as follows:

$$\begin{cases} ESR = \frac{R_{ac}}{(1 - \omega^2 L C_S)^2} \\ L_{eff} = \frac{L}{1 - \omega^2 L C_S} \end{cases} \tag{12}$$

When the relative dielectric constant of the concrete medium increases, the parasitic capacitance of the planar spiral coil will also increase; the parasitic capacitance will affect the self inductance and resistance of the planar spiral coil, as shown in Figure 5.

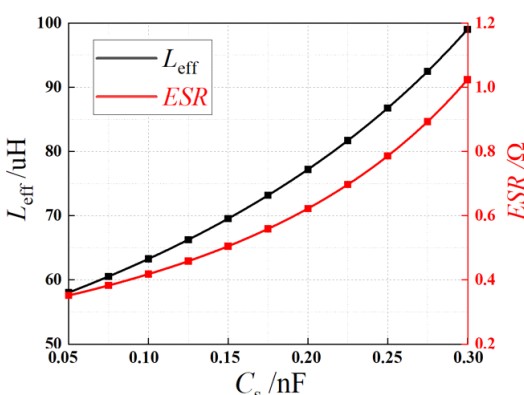

**Figure 5.** The curve of coil parameter change with the change of relative dielectric constant.

### 2.2. Eddy Current Loss Equivalent Resistance

Compared to air, concrete has a high electrical conductivity. In the case of concrete seepage, the charged particles in the water will increase the conductivity of the concrete, so the eddy current loss generated by the WPT system in the concrete needs to be considered. A cross-sectional view of the air–concrete cross-media WPT system is shown in Figure 6.

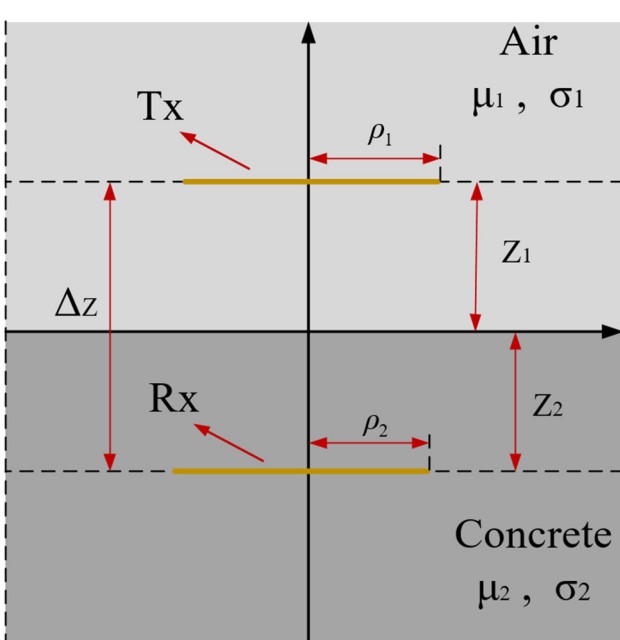

**Figure 6.** Cross section of air–concrete transmedia WPT system.

Equation (13) can be obtained using to Maxwell's equations [30]:

$$\begin{cases} \nabla \times H = J + \frac{\partial D}{\partial t} = (\sigma + j\omega\varepsilon)E \\ \nabla \times E = -\frac{\partial B}{\partial t} = -j\omega\mu H \\ \nabla \cdot B = 0 \\ \nabla \cdot D = 0 \end{cases} \tag{13}$$

After derivation, the Helmholtz equation can be obtained:

$$\nabla^2 E_i + k_i^2 E_i = 0, \, (k_i^2 = \omega^2 \mu_i \varepsilon_i - j\omega\mu_i \sigma_i) \tag{14}$$

Due to the spiral coil, the concrete is also simplified as a cylinder, so the electric field strength along the circumferential direction can be considered unchanged. Thus, the general solution for the electric field strength can be obtained as follows:

$$E_{it}(\rho, z) = \int_0^{+\infty} [C_1 J_1(\tau\rho) + C_2 N_1(\tau\rho)](C_3 e^{\sqrt{\tau_i^2 - k_i^2}z} + C_4 e^{-\sqrt{\tau_i^2 - k_i^2}z}) d\tau \quad (15)$$

where $J_1(x)$ is the first-order Bessel function and $N_1(x)$ is the first-order Neumann function. Since the first-order Neumann function is divergent, considering the boundedness of the electric field strength, $C_2 = 0$.

The boundary conditions of the electric field strength can be obtained as follows:

$$\begin{cases} \lim_{z \to -\infty} E_1 = 0 \\ \lim_{z \to +\infty} E_2 = 0 \\ \lim_{z \to 0+} E_2 - \lim_{z \to 0-} E_1 = 0 \\ \lim_{z \to 0+} \frac{\partial E_2}{\partial z} - \lim_{z \to 0-} \frac{\partial E_1}{\partial z} = j\omega u I \delta(\rho - a) \end{cases} \quad (16)$$

Substituting the boundary condition, Equation (17) can be obtained as follows:

$$\begin{cases} E_{1t,ii}(\rho, z) = -\frac{j\omega u I a_{ii}}{2} \int_0^{+\infty} \frac{\tau}{\sqrt{\tau_1^2 - k_1^2}} J_1(\tau a_{ii}) J_1(\tau\rho) e^{\sqrt{\tau_1^2 - k_1^2}z} d\tau, (d<0) \\ E_{2t,ii}(\rho, z) = -\frac{j\omega u I a_{ii}}{2} \int_0^{+\infty} \frac{\tau}{\sqrt{\tau_2^2 - k_2^2}} J_1(\tau a_{ii}) J_1(\tau\rho) e^{\sqrt{\tau_2^2 - k_2^2}z} d\tau, (d>0) \end{cases} \quad (17)$$

The total electric field strength of the coil with $N$ turns is shown in Equation (18):

$$E_{it} = \sum_{ii=1}^{N} E_{it,ii}(\rho, z) \quad (18)$$

The equivalent resistance of the spiral coil can be calculated as:

$$R_{\text{loss}} = \int_v \sigma \frac{|E_{it}|^2}{I^2} dV \quad (19)$$

The eddy current loss of the concrete medium can be equivalent to the resistance in the circuit, and the equivalent circuit model of the WPT system can be obtained, as shown in Figure 7.

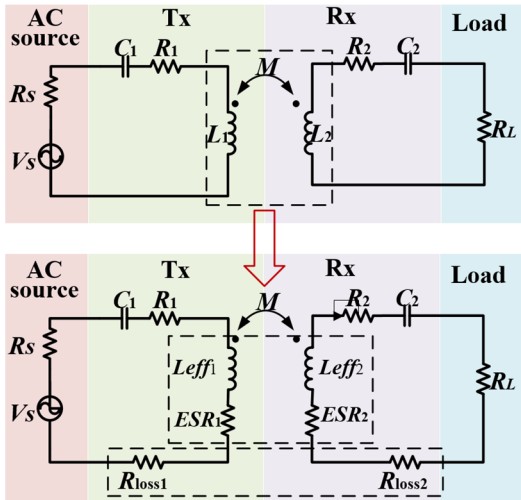

**Figure 7.** Equivalent circuit model of WPT system.

When the conductivity of the concrete medium changes, the eddy current loss induced by the transmitting coil will also change. At the same time, the value of the eddy current loss equivalent resistance in the equivalent circuit model will also change, as shown in Figure 8.

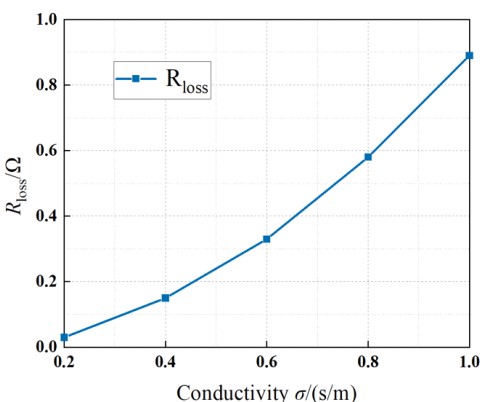

**Figure 8.** Curve of eddy current loss equivalent resistance change with conductivity change.

## 3. Analysis of Magnetic Coupling Characteristics

The equivalent circuit model of the air–concrete transmedia WPT system was established in the previous section. The equivalent circuit model elaborates the influence of the relative permittivity, conductivity and permeability of the concrete medium on the WPT system. In order to verify the correctness of the theory, it is necessary to simulate the air–concrete transmedia WPT system.

In this paper, the finite element software COMSOL 6.0 is used to establish a simulation model of the magnetic coupling mechanism. In order to facilitate the simulation, the planar spiral coil is equivalent to the ring model. At the same time, in order to verify the influence of the concrete medium on the resonant circuit, the magnetic field interface and the circuit interface are used to realize the coupling of the magnetic field and the electric field by setting the external circuit parameters to provide voltage for the coupling coil (i.e., external I vs. U interface), as shown in Figure 9.

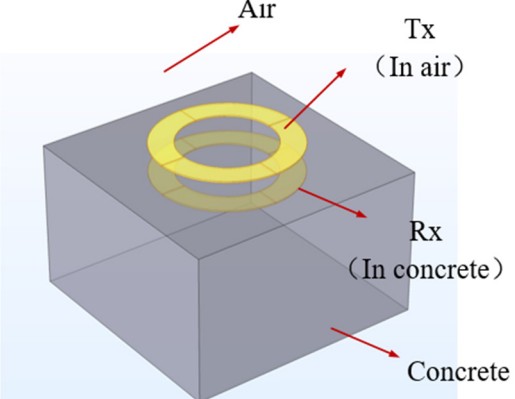

**Figure 9.** Simulation model and external circuit interface.

The model parameters are shown in Table 2.

**Table 2.** Simulation model parameters.

| Parameter | Value |
| --- | --- |
| Coil radius/mm | 20 |
| Coil diameter/mm | 2 |
| Coil turns | 10 |
| Excitation voltage size/V | 10 |
| Operating frequency/kHz | 85 |

In order to study the influence of concrete medium on WPT system, the simulation is carried out under the condition that the receiving coil is placed in air and the receiving coil is placed in concrete. The simulation parameters are set as follows: the relative dielectric constant of air is 1 and its conductivity is 0 S/m while the relative permittivity of concrete is 30 and its conductivity is 5 S/m. The simulation results are shown in Figure 10.

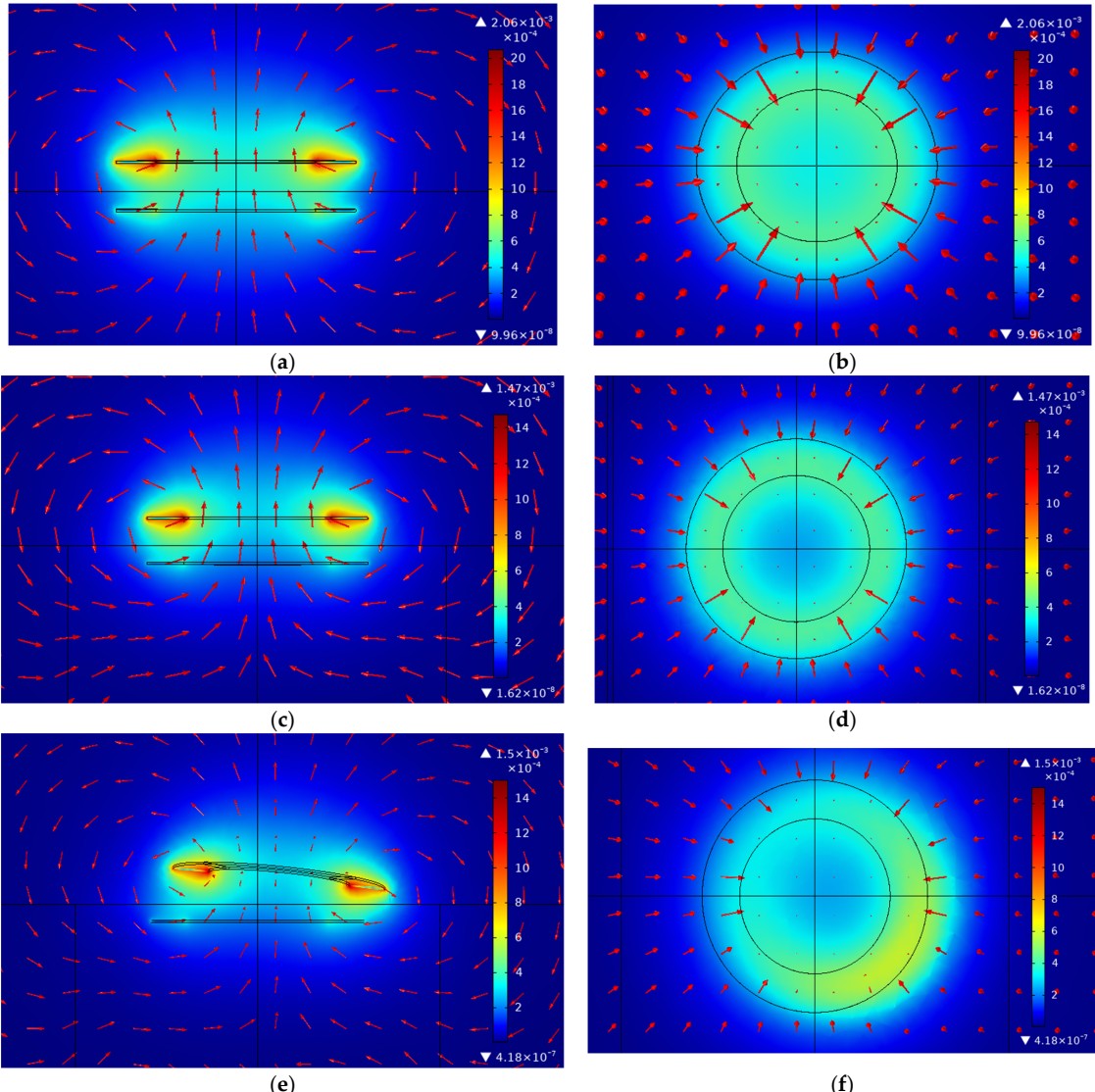

**Figure 10.** The distribution of the magnetic field before and after adding the concrete. Magnetic field of air: (**a**) longitudinal cross section and (**b**) transverse cross section. Magnetic field of concrete: (**c**) longitudinal cross section and (**d**) transverse cross section. Magnetic field distribution during migration: (**e**) longitudinal cross section and (**f**) transverse cross section.

It can be seen in Figure 10 that the magnetic field strength around the receiving coil in concrete decreases compared to the receiving coil in air. This indicates that the parasitic capacitance generated by the relative permittivity of the concrete medium affects the resonant circuit, so that the WPT system is out of the resonant state. At the same time, the high conductivity of the concrete medium also produces eddy current loss, which further weakens the magnetic field strength around the receiving coil.

Figure 11 illustrates the change of mutual inductance between the coupled coils of the WPT system when the coils are horizontally offset and angular deviated.

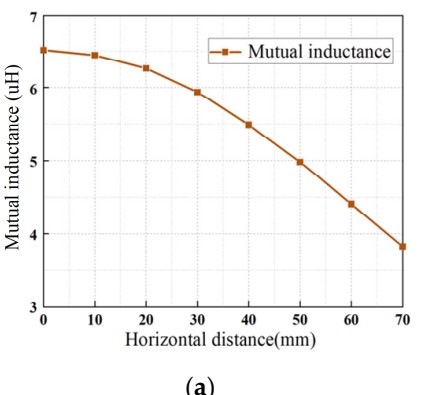
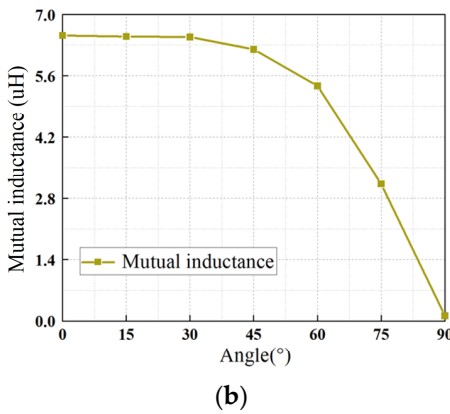

(**a**)                    (**b**)

**Figure 11.** Curve of mutual inductance change of WPT system: (**a**) mutual inductance variation curve with horizontal deviation distance and (**b**) mutual inductance varies with angle.

## 4. Tuning Control Strategy

In the last section, the simulation shows that under the influence of concrete medium, the intrinsic resonance frequencies of the primary and secondary side loops of the WPT system are shifted; this reduces the efficiency and output power of the WPT system. It is necessary to analyze the frequency characteristics of the WPT system before designing the tuning control strategy.

### 4.1. Frequency Response Analysis

The overall circuit model of the air–concrete transmedia WPT system is shown in Figure 6.

Without considering the concrete, for the S-S topology of the WPT system, the primary side loop and the secondary side loop satisfy Equation (20).

$$\begin{cases} Z_1 I_1 - j\omega M I_2 = V_S \\ -j\omega M I_1 + Z_2 I_2 = 0 \end{cases} \tag{20}$$

where $Z_1$ is the primary side impedance, $Z_2$ is the secondary side impedance.

$$\begin{cases} Z_1 = R_S + R_1 + j\omega L_1 + \frac{1}{j\omega C_1} \\ Z_2 = R_L + R_2 + j\omega L_2 + \frac{1}{j\omega C_2} \end{cases} \tag{21}$$

Ideally, when the primary and the secondary sides of the WPT system are completely resonant, the primary side impedance and the secondary side impedance are purely resistive, the system reactive power is 0 and the input power is completely converted into electrical energy and transmitted to the load. At this time, the output power and efficiency of the WPT system can be expressed as follows:

$$\begin{cases} P_{out} = \frac{\omega^2 M^2 V_s^2 R_L}{[(R_L+R_2)(R_S+R_1)+\omega^2 M^2]^2} \\ \eta = \frac{\omega^2 M^2 R_L}{(R_L+R_2)^2(R_S+R_1)+\omega^2 M^2(R_L+R_2)} \end{cases} \tag{22}$$

At this time, the output power and efficiency of the WPT system reach the maximum value; however, in the air–concrete cross-medium WPT system the WPT system is no longer in the resonant state due to the influence of the concrete medium. Because the receiving coil is placed in the air, the air has little effect on the inductance. For the convenience of analysis, it is assumed that the receiving coil inductance $L_1$ is constant and that the receiving coil inductance $L_2$ has changed. At this time, the output power and efficiency of the WPT system are described as follows:

$$
\begin{cases}
P_{out} = \dfrac{\omega^2 M^2 V_s^2 R_L}{\left[(R_L+R_2)^2+(R_S+R_1)+\omega^2 M^2\right]^2+\left[(R_S+R_1)(\omega\Delta L_2)\right]^2} \\
\eta = \dfrac{\omega^2 M^2 R_L}{\left[(R_L+R_2)^2+(\omega\Delta L_2)^2\right](R_S+R_1)+\omega^2 M^2(R_L+R_2)}
\end{cases}
\tag{23}
$$

where $\Delta L_2$ is the inductance variation of the receiving coil.

By comparing Equations (22) and (23), it can be found that the overall output efficiency of the WPT system is reduced due to the change of inductance, as shown in Figure 12.

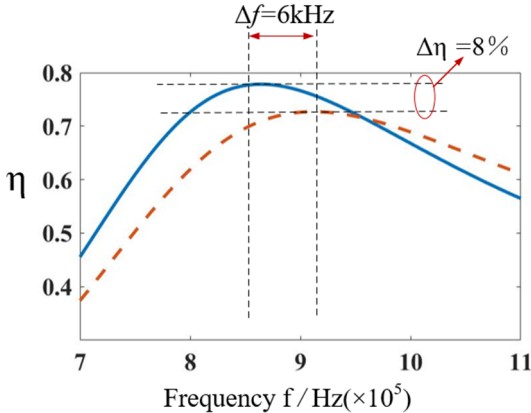

**Figure 12.** Efficiency curve of WPT system before and after detuning.

In Figure 12, it can be seen that after the detuning of the WPT system the efficiency of the WPT system is greatly reduced at the original operating frequency. The operating frequency of the system is an important factor affecting the transmission performance of the WPT system. In order to improve the efficiency of the WPT system, it is necessary to change the operating frequency of the system to reach the resonant state again after detuning.

### 4.2. Frequency-Modulation Tuning Control Strategy

Based on the frequency characteristics analyzed in Section 4.1, in the air–concrete cross-medium WPT system the concrete medium has an effect on the WPT system, which causes the WPT system to become detuned and thus affects the overall transmission performance of the system. In this paper, a phase-locked loop is used to tune the frequency modulation of a WPT system. The characteristic of a phase-locked loop is that the phase difference between the input signal and the output signal can be gradually changed to zero. When applied to a WPT system, the phase of the inverter output voltage and the primary current can be guaranteed to be consistent. Finally, the working frequency of the system is changed to improve the transmission performance of the WPT system.

The analog phase-locked loop is widely used currently; but, in recent years, the all-digital phase-locked loop has gradually replaced the analog phase-locked loop. Compared with the analog phase-locked loop, the digital phase-locked loop has the following advantages: simple circuit structure, strong filtering effect, less affected by input signal distortion and interference and not affected by the change of system operating frequency.

A phase-locked loop is essentially a closed-loop control loop; the principle is based on the difference between the input signal and the output signal to continuously reduce the phase difference between the two and ultimately realize the function of frequency tuning.

The core circuit of a phase-locked loop consists of three parts: p-phase detector, loop filter and voltage-controlled oscillator, as shown in Figure 13.

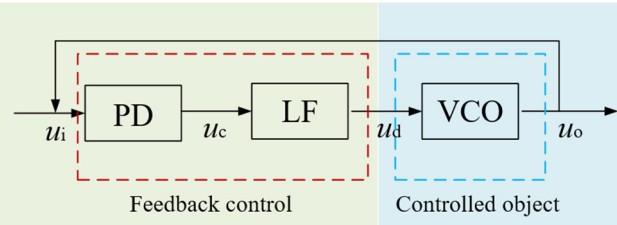

**Figure 13.** Basic schematic diagram of PLL. Where PD: phase detector, LF: loop filter and VCO: voltage-controlled oscillator.

The function of the phase detector is to compare the phase difference between the input signal and the output signal. Its internal model can be equivalent to the series connection of the multiplier and the low-pass filter. The input signal and the output signal are multiplied and are finally filtered by the low-pass filter. The phase difference signal is converted into a voltage signal and the transfer function is shown in Equation (24):

$$u_d(t) = K_d\theta \tag{24}$$

where $u_d$ is the output voltage signal of the phase detector, $K_d$ is the gain of the phase detector and $\theta$ is the phase difference between the input and output signals.

The function of the loop low-pass filter is to filter the output voltage signal of the phase detector. The output voltage of the phase detector also has a high-frequency noise interference signal. The loop filter can improve the transmission performance of the phase-locked loop circuit and greatly improve its stability. The transfer function is expressed in Equation (25):

$$U(s) = \frac{1}{1 + RCs} \tag{25}$$

The voltage-controlled oscillator is essentially an integral device. Its principle is to convert the voltage signal output via the loop low-pass filter and accumulate it to the frequency of the output signal, thereby changing the frequency of the output signal and changing the phase of the output signal according to the frequency of the output signal. Finally, the input signal and the output signal have no phase difference. The transfer function can be obtained as follows:

$$\omega_v(t) = \omega_o + K_0 u_c(t) \tag{26}$$

where $\omega_v$ is the instantaneous angular frequency of the voltage-controlled oscillator, $\omega_o$ is the inherent oscillation angular frequency of the voltage-controlled oscillator and $K_0$ is the sensitivity of the voltage-controlled oscillator.

In this paper, FPGA is used as the function of all-digital phase-locked loop. Figure 14 is the phase-locked loop control schematic diagram based on FPGA.

The control process is as follows: the output voltage and primary current of the inverter are collected and filtered and the voltage and current are converted into a 3.3 V square-wave signal by a zero-crossing comparator. Finally, the voltage and current square-wave signals are input into the FPGA for phase-locked loop frequency-modulation control and output PWM signal to drive the inverter.

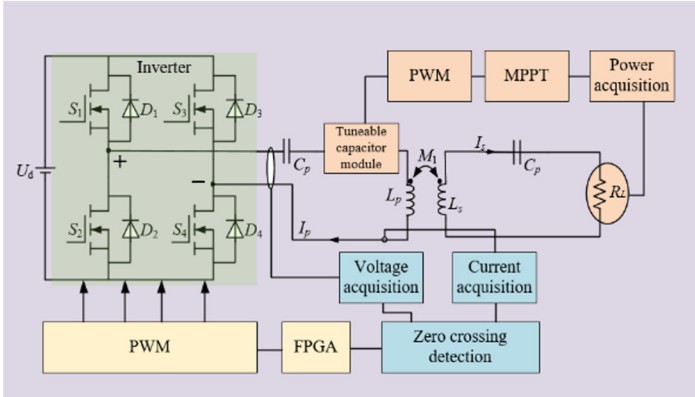

**Figure 14.** PLL control schematic diagram based on FPGA.

### 4.3. Dynamic Compensation Control Strategy

It can be seen from Section 2.1 that the relative permittivity of the external medium leads to parasitic capacitance of the coil of the WPT system and that the relative permittivity of the concrete medium is much larger than that of air. Therefore, the inherent resonance frequency offset of the transmitting coil placed in the air is small and the inherent resonance frequency offset of the receiving coil embedded in the concrete is large. According to Equation (23), when the secondary circuit is resonant, the efficiency of the WPT system with S-S topology reaches the maximum and the transmission performance of the WPT system reaches the optimal. The frequency-modulation tuning control strategy based on a phase-locked loop can only ensure that the WPT system is in the resonant state as a whole, but cannot ensure that the secondary circuit realizes resonance. Therefore, in this paper, a variable capacitor is added to the WPT system circuit for dynamic compensation and tuning control. Since the receiving coil is embedded in the concrete, it is difficult to replace and maintain it after adding a variable capacitor to the secondary circuit. The variable capacitor is composed of two MOS tubes in reverse series and a fixed capacitor in parallel [25]. The variable capacitor topology is shown in Figure 15.

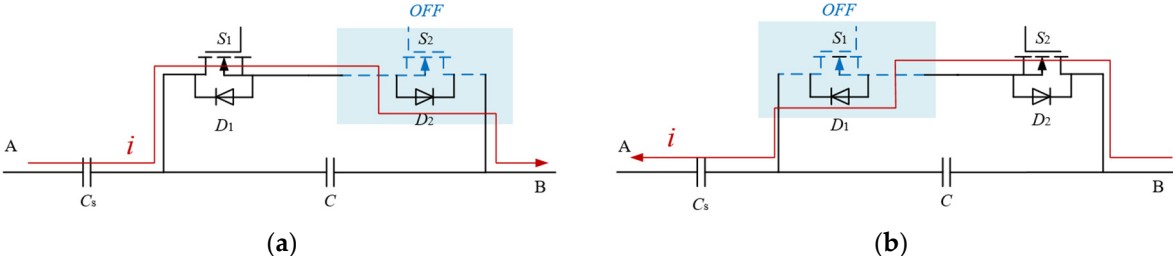

**Figure 15.** Topology of adjustable capacitance: (**a**) $i > 0$ and (**b**) $i < 0$.

The working principle of the variable capacitor is that when the input current $i_t$ is positive, the MOS tube $S_1$ is switched on. At this time, the input current $i_t$ does not flow through the fixed capacitor Ca and the voltage at both ends of the fixed capacitor is 0. At this moment, the waveform at both ends of the capacitor Ca can be changed and its equivalent capacitance value can be changed by controlling the conduction time of $S_1$ [29]. The voltage and current waveforms of the variable capacitor are shown in Figure 16. Its equivalent capacitance value is as follows:

$$C_{sc} = \frac{C}{2 - (2\alpha - \sin 2\alpha)/\pi} \tag{27}$$

$$C_{eq} = \frac{C_s C}{C + C_s[2 - (2\alpha - \sin 2\alpha)/\pi]} \tag{28}$$

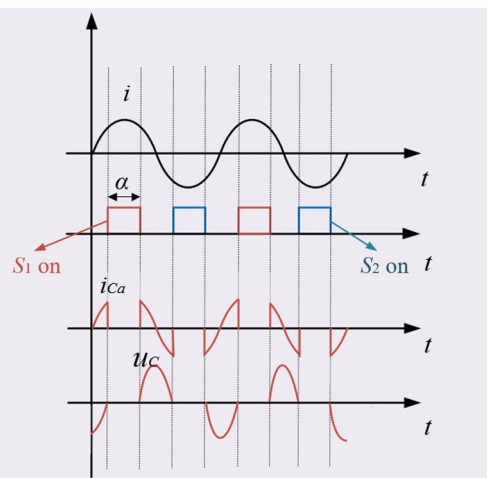

**Figure 16.** Adjustable capacitor voltage-current waveform. Where $i$ is adjustable capacitor module input current, $\alpha$ is the on-angle of the MOS tube, $i_{Ca}$ is the current flowing through capacitor $C$, $u_c$ is the voltage of capacitor $C$.

## 5. Experimental Verification

In order to verify the effectiveness and feasibility of the control strategy, an experimental prototype of WPT system with compound tuning control strategy is designed and built in this paper, as shown in Figure 17.

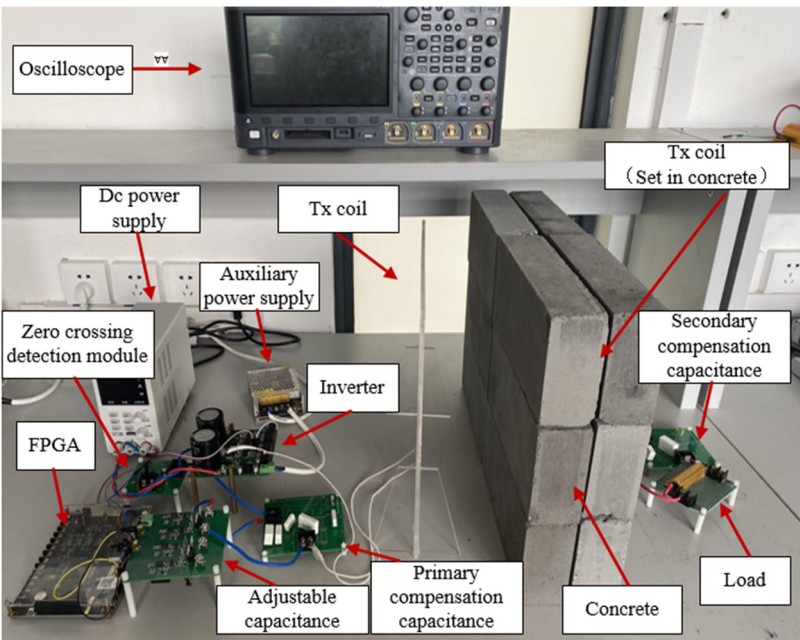

**Figure 17.** The experimental prototype.

In Figure 17, the transmitting coil is placed in the air, the receiving coil is placed in the concrete brick and the wire is drawn to provide power for the load. The receiving coil and the transmitting coil are both planar spiral coils, the turns are both 10 turns and the coil height radius is 14 mm. The system parameters are shown in Table 3.

**Table 3.** Experimental equipment parameter.

| Parameter | Value |
|---|---|
| DC voltage source E/V | 12 |
| Initial system frequency f/kHz | 85 |
| The distance between the primary and secondary coil is l/mm. | 100 |
| Load resistance $R_{load}/\Omega$ | 5 |
| Transmitting coil inductance $L_1/\mu H$ | 53.8 |
| Primary compensation capacitor $C_p/nF$ | 65.49 |
| Receiving coil inductance $L_1/\mu H$ | 53.6 |
| Primary compensation capacitor $C_p/nF$ | 65.52 |

*5.1. Analysis of System Power Transmission Characteristics*

The output voltage and primary current waveforms of the inverter are shown in Figure 18 after the system is powered on and before the tuning control is performed.

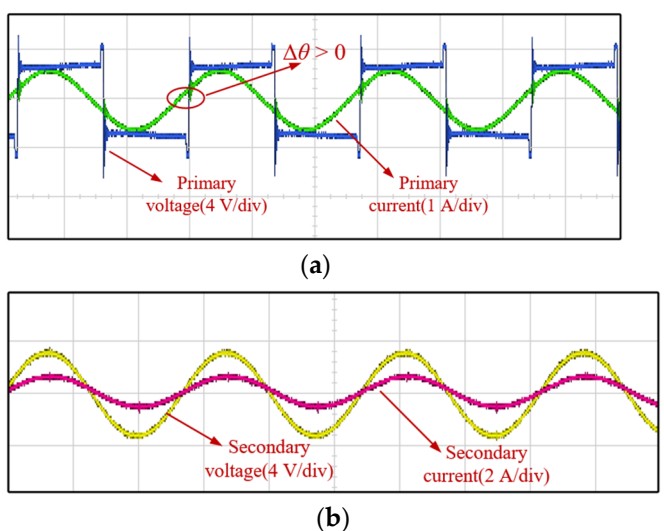

(**a**)

(**b**)

**Figure 18.** Inverter output voltage and primary current waveform: (**a**) primary voltage and current waveform and (**b**) secondary voltage and current waveform.

It can be seen in Figure 19 that, before the tuning control, the inverter output voltage and the primary current have a significant phase difference $\Delta\theta$. The primary current leads to the inverter output voltage, which indicates that the concrete medium has an effect on the coupling mechanism, making the WPT system deviate from the resonant state. The primary current is 0.56 A. The load voltage is 2.84 V and the load current is 0.53 A. The output power of the system is 1.51 W and the efficiency is 71.63%.

After adding tuning control, the output voltage and primary current waveforms of the inverter are shown in Figure 16. At this time, the primary current is 0.96 A, the load voltage is 3.67 V and the load current is 0.71 A. The output power of the system is 2.61 W and the efficiency is 82.68%. Experimental data before and after tuning are shown in Table 4.

**Table 4.** Experimental result.

| | Before Tuning Control | After Tuning Control |
|---|---|---|
| Primary voltage/V | 3.62 | 3.39 |
| Primary current/A | 0.56 | 0.96 |
| Secondary voltage/V | 2.84 | 3.67 |
| Secondary current/A | 0.53 | 0.71 |
| Output power/W | 1.51 | 2.61 |
| Efficiency | 71.63% | 82.68% |

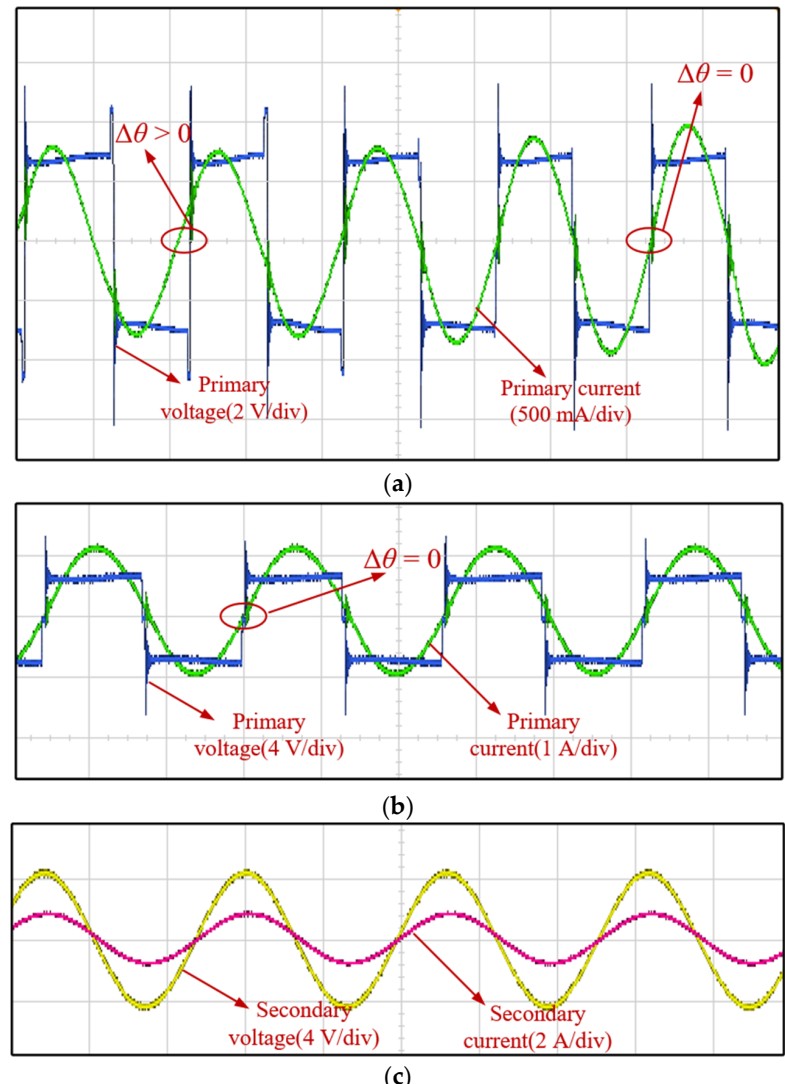

**Figure 19.** The output voltage and primary current waveform of the inverter: (**a**) in the tuning process, (**b**) primary voltage and current waveform after tuning and (**c**) secondary voltage and current waveform after tuning.

The experimental results are compared with other references, as shown in Table 5.

**Table 5.** Comparative analysis.

| Research | Optimization Mode | Control Link | Application Situation | Efficiency Improvement |
|---|---|---|---|---|
| Reference [19] | × | Open loop | Transportation | / |
| Reference [24] | × | Open loop | Underwater equipment | / |
| Reference [32] | Coil optimization | Open loop | Underwater equipment | 10% |
| Reference [33] | Compensating inductance | Open loop | Underwater equipment | 7.3% |
| Reference [34] | Controlled rectification | Closed loop | / | 6.58% |
| This paper | Digital phase-locked loop | Closed loop | Concrete | 11.05% |

where "×" means that no relevant studies have been conducted, "/" means that nothing relevant was mentioned.

### 5.2. Analysis of System Frequency Characteristics

Before and after the tuning process of the system, the operating frequency of the WPT system measured by the spectrum analyzer waveform is shown in Figure 20.

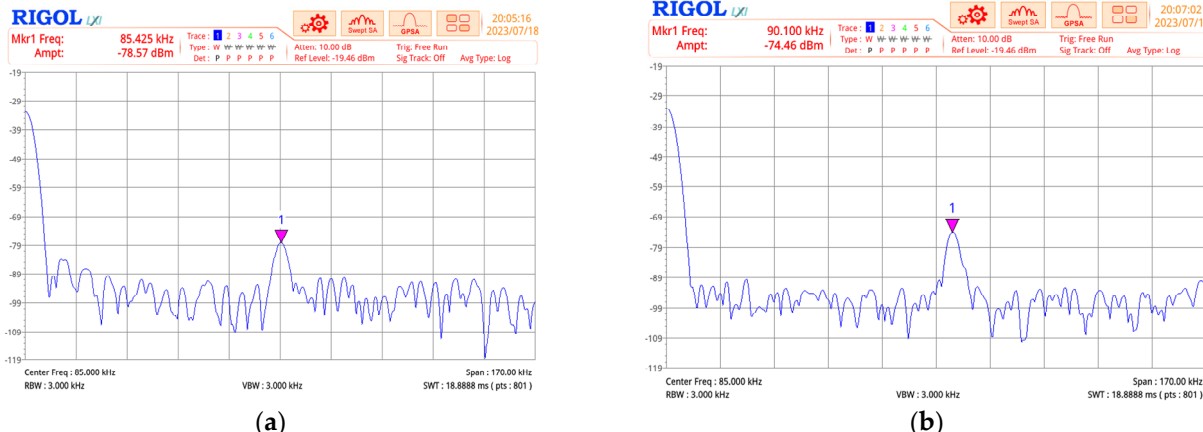

**Figure 20.** Spectrum waveform of WPT system: (**a**) before tuning and (**b**) after tuning.

It can be seen in Figure 20 that the operating frequency of the WPT system is about 85 kHz before tuning. After the tuning is completed, the operating frequency of the system becomes about 90 kHz and the amplitude is improved. The spectrum waveform in the tuning process is shown in Figure 21.

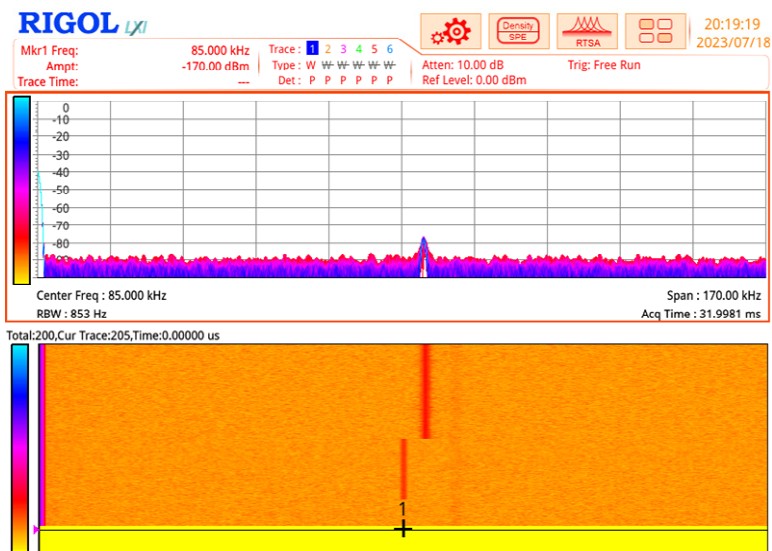

**Figure 21.** Spectrum waveform of WPT system during tuning.

### 6. Conclusions

This paper proposes a composite tuning control technology based on the combination of frequency-modulation tuning and dynamic compensation tuning for concrete embedded sensors' wireless charging. By using this tuning control technology, the reduced power transmission efficiency caused by system detuning can be effectively improved. At the same time, the frequency characteristics of the WPT system are also analyzed and compared. Finally, experimental verification was conducted and the results showed that the system output power and efficiency were significantly improved, with an output power increase of 73% and an efficiency increase of 11.05% after adopting tuning control. In addition, the operating frequency of the system has been increased from 85 kHz to 90 kHz. Therefore, it

has been verified that the proposed tuning control technology has the ability to improve the energy transfer efficiency of the air–concrete cross-dielectric WPT system.

**Author Contributions:** Conceptualization, C.R.; Methodology, C.R. and Z.W.; Software, M.C.; Investigation, C.X.; Resources, C.R.; Data curation, J.Y. and G.R.; Writing—review & editing, L.Y.; Supervision, C.R. All authors have read and agreed to the published version of the manuscript.

**Funding:** This work was supported by the State Key Laboratory of Advanced Electromagnetic Engineering and Technology under Grant AEET 2022KF007.

**Data Availability Statement:** Not applicable.

**Conflicts of Interest:** The authors declare no conflict of interest.

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
