# Peer review of "Research on Tuning Control Technology for Wireless Power Transfer Systems for Concrete Embedded Sensors"

_electronics, doi:10.3390/electronics12183963_

Round 1
Reviewer 1 Report
The reviewed article is devoted to a rather interesting and new direction in modern construction industry related to IoT use in creating a new generation of "smart" concrete and other cement-based construction materials with built-in condition monitoring sensors. Therefore, the increasing efficiency of WPT technology for charging sensors embedded in the concrete is also relevant. According to the data given in the article, the tuning control proposed by authors allowed to increase the efficiency of energy transmission by more than 11%.
As a main comment, I want to note that Section 1, where mathematical descriptions of equivalent schemes are given, is declarative in nature. Why are these models given in the text of the article? The supplementing sections 1.1. and 1.2 with some conclusions or analysis of the expressions obtained may be the worth.
There are also some formal remarks. The authors should indicate the abbreviations used in the figure 10 after their specifying in lines 300-304. Fig.11 contains English and Chinese designations.
However, my remarks are not essential, and the article can be published after minor improvements.
Author Response
To: Electronics Editor
Re: Response to reviewers
Dear Editor,
Thank you for your hard work.
The authors of this manuscript greatly appreciate the valuable comments from the reviewers. The suggestions are greatly helpful to improve the quality of our manuscript. We have made careful modification strictly according to the reviewers’ advices which we hope meet with approval. We are uploading (a) our point-by-point response to the comments (below) (response to reviewers), (b) an updated manuscript with yellow highlighting indicating changes, and (c) a clean updated manuscript without highlights (PDF main document).
Best regards,
Cancan Rong, (First author& Corresponding author)

Reviewer 2 Report
In the present article, a composite tuning control technology based on the combination of frequency modulation adjustment and dynamic compensation adjustment was proposed based on the influence of concrete on the WPT system coil in the WPT air-concrete cross-medium system, which leads to the detuning of the WPT system. Thus, through the theoretical analysis of the electrical parameters of the concrete medium, this article finds that the concrete medium causes the inductance of the coil to change, the receiver coil built into the concrete is most affected, and the final transmission efficiency is greatly reduced. Basis, the article designs a control system that combines frequency tracking and dynamics compensation to improve the output power and efficiency of the WPT system under the influence of concrete. The experimental results show that the transmission performance of the WPT system was greatly improved after adopting the tuning control strategy.
Therefore, the problem of low efficiency caused by resonance frequency misalignment due to the influence of the concrete medium on the air-concrete cross-medium WPT system is sorted out. The article is well written and well reasoned with new contributions. Therefore, the article is accepted for publication.
Author Response
Thank you for your hard work.
The authors of this manuscript greatly appreciate the valuable comments from the reviewers. The suggestions are greatly helpful to improve the quality of our manuscript. We have made careful modification strictly according to the reviewers’ advices which we hope meet with approval. We are uploading (a) our point-by-point response to the comments (below) (response to reviewers), (b) an updated manuscript with yellow highlighting indicating changes, and (c) a clean updated manuscript without highlights (PDF main document).
Reviewer 3 Report
This work is promising, but it can be further improved by emphasizing the methodology and novel contributions in both the abstract and conclusion.
Author Response

(The authors gave the same response as above.)

Reviewer 4 Report
a composite tuning control technology combining frequency modulation tun16 ing and dynamic compensation tuning is proposed in this paper. Then, the loss of WPT system in concrete medium is analyzed, and the induced dielectric loss and eddy current loss in concrete are calculated by analytical equation and finite element analysis method. Subsequently, the equival.
I think the article is well written and the idea is good, but the authors need to improve.
1- the motivations in the introduction
2- I think that other articles are needed in the literature and an in-depth comparison should be added in the form of a table.
3- it is recommended in particular to add a table to describe the parameters
4-the results should be compared with the literature
5- the conclusion needs a second reading
moderate english is required
Author Response

(The authors gave the same response as above.)

Reviewer 5 Report
Review of article:
Research on tuning control technology of WPT system for concrete embedded sensors
The authors describe the problem of concrete-embedded receiver coils for embedded sensors. Due to the influence of the concrete, the inductance of the receiver coil can change which can detune the system, leading to lower system efficiency. The authors provide the mathematical model of the system and how change in inductance affects the overall system efficiency. The proposed method for increasing the efficiency consists of frequency tuning and tuning of the resonant capacitor. The proposed control strategy was verified experimentally.
Comments:
- Line 31: Introduction should be numbered as 1 and not 0.
- Line 156: Equation has a symbolic error.
- Line 168: Equation has a symbolic error.
- Line 182: Equation has a symbolic error.
- Figure 11 includes some Chinese symbols. Please correct that.
- Figure 11 should also include the tuneable capacitor for dynamic compensation.
- The proposed PLL ensures that the phase difference between the input current and input voltage is zero, as can be seen in experimental results. However, IPT systems are usually excited above the resonant frequency to ensure soft-switching. What are the soft-switching capabilities of your proposed control scheme?
- Section 3.3 has missing references to the tuneable capacitors. Also, the equation describing the connection between the pulse width α and the equivalent capacitance should be added.
- What is the algorithm for tuning the tuneable capacitor in connection to the PLL control strategy?
- The control described in the paper only ensures optimal and most efficient power transfer frequency. The current/voltage control of the receiver side was not described or mentioned.
- The authors should also include the results, of how coil misalignment affects the efficiency of the system. Does this also affect the phase angle between the input current and voltage?

Author Response

(The authors gave the same response as above.)

Round 2
Reviewer 5 Report
The author responded to all reviewers questions. There are no additional comments.